# A query engine for L1-L2 parallel dependency treebanks

**Arianna Masciolini**

Språkbanken Text
Department of Swedish, Multilingualism, Language Technology
University of Gothenburg
`arianna.masciolini@gu.se`

## Abstract

L1-L2 parallel dependency treebanks are learner corpora with interoperability as their main design goal. They consist of sentences produced by learners of a second language (L2) paired with native-like (L1) correction hypotheses. Rather than explicitly labelled for errors, these are annotated following the Universal Dependencies standard. This implies relying on tree queries for error retrieval. Work in this direction is, however, limited. We present a query engine for L1-L2 treebanks and evaluate it on two corpora, one manually validated and one automatically parsed.

## 1 Introduction

L1-L2 parallel dependency treebanks are learner corpora where sentences produced by learners of a second language (L2) are paired with correction hypotheses, assumed to be native-like and therefore referred to as *L1 sentences*. Both the learner originals and the corresponding corrections are annotated following the cross-lingual Universal Dependencies (UD) standard (Nivre et al., 2020). The idea is that such morphosyntactical information makes explicit error labelling unnecessary and allows errors to instead be retrieved via tree queries. This format, proposed by Lee et al. (2017a), was in fact designed to address the interoperability issues arising from the coexistence of the different markup styles and error taxonomies normally employed for the annotation of learner corpora. These tend not only to be language-specific, but also to vary widely across different same-language projects. An additional advantage of using UD is the availability of several increasingly fast and reliable parsers (Straka, 2018; Qi et al., 2020). While not yet very robust to learner errors (Huang et al., 2018), they can already speed up the annotation process significantly.

L1-L2 UD treebanks exist for English (Berzak et al., 2016), Chinese (Lee et al., 2017b) and Italian (Di Nuovo et al., 2022). Work on error retrieval tools, on the other hand, has been limited. Only one of these corpora, the ESL (English as a Second Language) treebank, is equipped with a query engine.[1] This tool, however, presents several limitations, a major one being its reliance on a pre-existing error taxonomy, in contrast with Lee et al. (2017a)'s idea.[2] Closer in spirit to the latter, Choshen et al. (2020) have developed a method to automatically derive dynamic syntactical error taxonomies from L1-L2 treebanks, but do not provide a way to look for specific error patterns.

In this paper, we present a language- and error taxonomy-agnostic query engine for L1-L2 parallel dependency treebanks. The tool allows searching for morphosyntactical errors by describing them in a pre-existing pattern matching language for UD trees, which we extend to facilitate comparing L2 sentences to their corrections, resulting in what we call *L1-L2 patterns*. Our main contribution is a sentence retrieval algorithm that matches the L1 and L2 portions of a query pattern on the corresponding treebanks in parallel, ensuring that correspondences are found between segments that align with each other. Addressing another limitation of the existing tools, we also make it possible to extract the specific portions of an L1-L2 sentence that match a given pattern. The engine is part of L2-UD, a larger open source toolkit for UD-annotated L2 data, available for download at `github.com/harisont/L2-UD`.[3]

---

[1]As of 05.04.2023, the ESL treebank's homepage, `esltreebank.org`, seems to be no longer reachable, but the user interface of the query engine can be inspected at the Internet Archive: `web.archive.org/web/20220120204838/http://esltreebank.org`.

[2]Note, however, that the ESL treebank actually predates Lee et al.'s paper.

[3]The results reported in this paper were obtained with version 0 of the engine: `github.com/harisont/L2-UD/releases/tag/v0` (last access 05.04.2023).

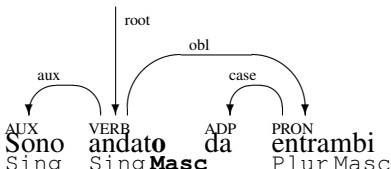 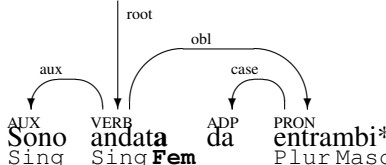

Figure 1: UD trees for the Italian sentence *Sono andat{o→a\*} da entrambi* ("I have been to both"), discrepancies highlighted in bold. In the L2 sentence, displayed on the right, the gender of the participle *andato*, referring to the implicit subject of the sentence, is incorrect, but without further context we have no way to infer the author's gender. As a consequence, the error can only be described in terms of its correction. Example adapted from the VALICO-UD treebank.

## 2 Related work

As mentioned in the introduction, learner corpora exist in a variety of formats. Lee et al. (2017a)'s proposal to use L1-L2 parallel dependency treebanks is not the only one aimed at overcoming the interoperability issues that follow. Bryant et al. (2017), for instance, introduced ERRANT, an ERRor ANnotation Toolkit operating in the framework of a taxonomy that exclusively relies on dataset-agnostic information such as the POS (Part Of Speech) tag and morphological features of the tokens involved. In a sense, this can be seen as an attempt to solve the problem by developing a "universal" error taxonomy. While ERRANT has become dominant in Grammatical Error Correction research, it still coexists with several other tagsets which differ significantly both in their underlying assumptions, often language-specific, and in the granularity of the annotation, which varies according to the intended use of each individual corpus.

Lee et al.'s idea, while only concerned with morphosyntactical errors, is more radical, as UD-annotated parallel treebanks have the potential to remove the need for any explicit error pre-categorization and instead allow to infer error taxonomies automatically and dynamically. Choshen et al.'s work on syntactical error classification, showing promising results even on automatically parsed L1-L2 treebanks, goes in this direction.

While there is a wide variety of tools and language to choose from for extracting information from (monolingual) UD treebanks,[4] not many options are available when it comes to retrieving example sentences matching specific patterns of error from L1-L2 treebanks. To the best of our knowledge, the above mentioned ESL treebank query engine is the only tool specifically meant for this task. While it is reasonable to assume that the latter could easily be generalized to work

with any L1-L2 treebank, it presents several limitations from the perspective of the tree queries envisioned by Lee et al. (2017a). First and foremost, searching for errors is primarily done by selecting an error label from a pre-defined set. A simple query language is also available, but it only allows searching for sequences of word forms, POS tags and dependency labels. In other words, UD sentences are treated as lists of tokens rather than trees. This can be restrictive since, for the purposes of grammatical error retrieval, dependency structure is often more relevant than linear order. Furthermore, there is no coupling, other than sentence-level alignment, between the L1 and L2 parts of the treebank. Patterns are therefore only matched against L2 sentences, making it impossible to search for errors whose description requires a comparison with the correction (cf. Figure 1 for an example) or locate the relevant portions of their L1 counterparts. A final, related limitation is that the tool always returns complete sentences, while it is sometimes useful to isolate the segments that match the query.

## 3 Design and implementation

Addressing these limitations, we aim for a query engine with the following characteristics:

1. **no underlying error taxonomy**: errors are described in a pattern matching language which allows treating UD sentences both as sequences of tokens and as tree structures;
2. **parallel L1-L2 matching**: queries consist in an L1 pattern, that has to be matched by a correction hypothesis, and an L2 pattern to be matched in the corresponding learner sentence. This also allows formulating queries by comparing learner sentences with their corrections;
3. **subsentence extraction**: besides retrieving full sentence pairs, it is also possible to extract the specific portions of an L1-L2 pair actually matching the query.

---

[4]PML-TQ (Pajas and Štěpánek, 2009), GREW-MATCH (Guillaume, 2021), SETS (Luotolahti et al., 2015) and TÜNDRA (Martens, 2013), just to name a few.

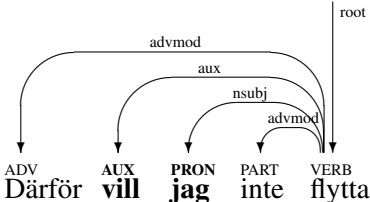 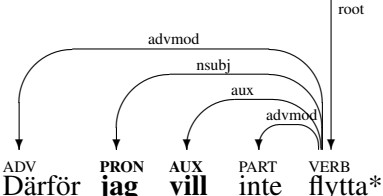

Figure 2: UD trees for the L1-L2 Swedish sentence *Därför {vill jag → jag vill*} inte flytta* ("Therefore I don't want to move"), discrepancies highlighted in bold. The L2 sentence, on the right, violates V2 word order. Example adapted from the DaLAJ corpus.

## 3.1 Query language

A central part of the engine is its query language. We start with an overview of the pre-existing pattern matching language our system makes use of. After that, we present the extensions through which we adapt it to querying L1-L2 treebanks.

### 3.1.1 UD patterns

To describe morphosyntactical structures, we use the Haskell-embedded pattern matching language available as part of the GF-UD toolset for dependency trees and interlingual syntax (Kolachina and Ranta, 2016; Ranta and Kolachina, 2017).[5] Although not as widespread as some of the above mentioned alternatives, it allows to express a wide range of queries with an intuitive syntax, and it was selected due to its ease of integration with the other components of the project.

In essence, the language provides three types of patterns:

- *single-token patterns*, e.g. `POS "VERB"`, matching all (sub)trees rooted in a verb. With a similar syntax, it is possible to pattern match based on the token's `XPOS`, `DEPREL`, `FEATS`, `FORM` or `LEMMA`, all of which correspond to homonymous CoNNL-U fields;[6]
- *tree patterns*, in the form `TREE p [ps]`, where `p` is a pattern to be matched by the root node and `[ps]` an ordered list of patterns denoting its dependents. For instance, the pattern `TREE (POS "VERB") [DEPREL "nsubj", DEPREL "obj"]` matches all subtrees rooted in a verb having exactly two subtrees: a nominal subject `nsubj` and a direct object `obj`, in this order;

---

[5] For an exhaustive description of the pattern matching language, see the relevant GF-UD documentation: github.com/GrammaticalFramework/gf-ud/blob/master/doc/patterns.md (last access 05.04.2023).

[6] For an overview of the CoNNL-U format and a complete list of the abbreviations used in this text, see Appendix A.

- *sequence patterns*, matching subtrees where a certain sequence of patterns occurs with no intervening words. For instance, in Subject-Verb-Object (SVO) languages we might want to write `SEQUENCE [POS "VERB", DEPREL "nsubj", DEPREL "obj"]`.

More liberal versions of some of these patterns, using the original name followed by an underscore, also exists. Namely, `DEPREL_ d` ignores relation subtypes, `FEATS_ fs` matches all tokens whose morphological features include (rather than conicide with) `fs`, `TREE_ p [ps]` allows other dependents to appear before, between and/or after the explicitly listed ones and `SEQUENCE_ ps` does not require the listed patterns to occur contiguously. In addition, the language allows to combine patterns with the logical operators `AND`, `OR` and `NOT` and provides a `TRUE` pattern matching any subtree.

As a slightly more complex example, consider

```
TREE_
  (POS "VERB")
  [DEPREL_ "nsubj",
   OR [DEPREL "obj", DEPREL "obl"]]
```

The above pattern matches any subtree rooted in a verb which has at least two dependents: a nominal subject (ignoring any subtyping) and a direct object `obj` or oblique `obl` (not subtyped).

### 3.1.2 L1-L2 patterns

In some cases, errors can be described by a single UD pattern to be looked for in the L2 treebank. Often, however, it is more convenient and concise (if not even necessary, as illustrated in Figure 1) to describe errors by comparing an L2 sentence to its correction. For this reason, queries in our system are defined as pairs of UD patterns. This, however, does not prevent writing queries as L2-only patterns: any single-pattern query `q` is simply expanded to a pair ⟨TRUE, q⟩.

As an example of the usefulness of L1-L2 patterns, consider the sentence displayed in Figure 2: in the L2 text displayed on the right, the learner is

using Swedish's default SVO order, with the pronoun *jag* preceding the auxiliary verb *vill*. The sentence, however, starts with the adverb *därför*. Being Swedish a language with verb-second (V2) word order, the correction, displayed on the left, moves the auxiliary in the second position, right after the adverb itself. A way to find L2 sentences presenting the same problem is to use a single sequence pattern, for instance:

```
SEQUENCE [
  POS "ADV",
  OR [POS "VERB", POS "AUX"],
  DEPREL_ "nsubj"]
```

This does match sentences like the one above, but does not cover all cases in which V2 order is violated: rather than with an adverb, the sentence might for example start with a prepositional phrase (cf. *På grund av detta vill jag inte flytta*, with the similar meaning of "Because of this I don't want to move"). Rather than enumerating all possible patterns of V2 order violation, it can be more convenient to express the error in terms of its correction, for instance with the following pair of patterns, which disregards the opening phrase:

```
L1 : SEQUENCE [
       OR [POS "VERB", POS "AUX"],
       DEPREL "nsubj"]
L2 : SEQUENCE [
       DEPREL "nsubj",
       OR [POS "VERB", POS "AUX"]]
```

For the sake of conciseness, rather than writing two separate patterns, we enclose the discrepant portion in curly brackets and divide the L1 and L2 segments with an arrow:

```
SEQUENCE [
  {OR [POS "VERB", POS "AUX"],
  DEPREL "nsubj" →
  DEPREL "nsubj",
  OR [POS "VERB", POS "AUX"]}]
```

This is our first extension to the pattern matching language described in Section 3.1.1.

To avoid repetition, we also introduce variables. As an example use case, consider gender agreement, a source af confusion for learners of many languages. In a dependency tree, most errors of this kind can be identified by checking whether the gender of certain dependents matches the gender of the token they are referred to. In Italian, for instance, adjectives should agree with the nouns they modify. This is not the case in the sentences like *Indossava una maglietta nero* ("(S)he was wearing a black t-shirt"), where the noun, *maglietta*, is feminine, while the adjective is incorrectly in-

flected in its masculine form *nero*. This particular sentence therefore matches the pattern

```
TREE_
  (FEATS_ "Gender=Fem")
  [AND [DEPREL "amod",
        FEATS_ "Gender=Masc"]]
```

With the syntax presented until now, however, looking for all noun-adjective gender agreement errors requires a separate query for each possible combination of genders.[7] With variables, syntactically characterized by capital letters preceded by a `$` sign, we can instead simply write

```
TREE_
  (FEATS_ "Gender=$A")
  [AND [DEPREL "amod",
        FEATS_ "Gender=$B"]]
```

where `$A` is assumed to be different from `$B`. Variables are currently supported for morphological features, Universal POS tags and dependency relations, all of which have a finite number of possible values.

## 3.2 Sentence retrieval algorithm

Alongside the pattern matching language, GF-UD provides a function that, given a pattern and a UD tree, recursively checks if the former matches the latter itself or any of its subtrees. One might be prone to think that performing an L1-L2 query can simply consist in applying this function to all trees in the treebank, looking for L1 sentences matching the L1 portion of the pattern and L2 sentences matching its L2 portion. Doing that, however, generally leads to a significant amount of false positives. Consider, for instance, the following query, intended for searching number agreement errors between a head and its direct dependents:

```
TREE_
  (FEATS_ "Number=$A")
  [FEATS_ "Number={$A → $B}"]
```

Following this naïve approach, the sentence in Figure 1 would match the pattern even if it does not contain a number agreement error. This happens because the L1 sentence matches the L1 pattern at *sono andato* ("(I) have been", two singular verb forms), while the L2 sentence matches the L2 pattern at *andata da entrambi* ("been to both"), where the head *andata* is again a singular but the dependent *entrambi* is a pronoun in its masculine plural form. In this case, in fact, both the original

---

[7]Two in Italian, whose only genders are masculine and feminine, but already six for languages with neuter!

sentence and its correction match both portions of the pattern.

A key observation here is that *sono andato* and *andata da entrambi* do not semantically correspond to each other: to solve the problem, we need to further align our L1-L2 treebank, recursively putting L1 subtrees in correspondence with their L2 counterparts. To do that, we use the CONCEPT-ALIGNMENT Haskell library (Masciolini and Ranta, 2021). While originally designed for extracting translation equivalents from multilingual parallel treebanks, its *alignment criteria*, i.e. the set of rules to decide whether two subtrees correspond to each other, are configurable and easy to adapt to the L1-L2 domain. Actually, accuracy on L1-L2 corpora tends to be better than it is for multilingual treebanks. Learner sentences and their corrections, in fact, usually share the vast majority of the lemmas, something that can be taken into account when defining custom alignment criteria.

As a first step, then, we extract phrase- and word alignments, in the form of pairs of L1-L2 UD trees, for each L1-L2 sentence pair. After that, to decide whether a sentence pair matches a given L1-L2 pattern, we apply a nonrecursive version of GF-UD's pattern matching function to check if there is a pair of aligned subtrees whose L1 (resp. L2) component matches the L1 (resp. L2) portion of the pattern. Matching nonrecursively, only on complete UD trees (although extracted from full sentences), is crucial here, as it is, in most cases, what prevents L1-L2 patterns from being matched in structurally similar but semantically unrelated subtrees of the L1-L2 sentence pair.

The careful reader, however, will have noticed that this does not solve the issue for the specific example mentioned, where the subtrees matching the L1-L2 pattern, *sono andata* and *andata da entrambi*, share the same head *andata*. The false positive is due to the fact that its dependents, *sono* and *da entrambi*, do not correspond to each other. For TREE and TREE_ patterns, then, we recursively perform the additional check that all dependents involved in the match are aligned with each other. A similar mechanism is in place for SEQUENCE and SEQUENCE_ patterns, to avoid matching subsequences that, while part of the same subtree, are not semantically equivalent.

## 3.3 Subsentence extraction

By default, the output of the program is the list of IDs of the sentences matching the given query. Nontheless, extracting relevant subsentences can be useful both for futher processing of the error-correction pair and to more easily visualize discrepancies in the context in which they occur.

The fact that our sentence retrieval algorithm applies patterns on sub-sentence alignments makes it straightforward to locate the specific L1 and L2 subtrees where the match is found. Doing so, however, is of very limited usefulness when the root (or the head of a large subtree) is involved in the error, resulting in too big subtree pairs. For this reason, we prune the extracted subtrees by only keeping the portions explicitly described by the pattern: individual heads for single-token queries, heads and their dependents that match a pattern in ts for TREE_ ts patterns and, for sequence patterns, rather than the whole subtree including the given sequence, only subtrees matching one of patterns explicitly listed in it. Implementation-wise, this is done by converting the query's UD patterns into *replacement patterns* in GF-UD's tree manipulation language.[8]

The engine has options to either extract such pairs of matching subsentences and write them to CoNNL-U files or to output a Markdown report where they are highlighted in the sentences where they occur. Example reports obtained with the latter method can be found in Appendix C.

## 4 Evaluation

Aiming at assessing the performance of the query engine, we tested it on two L1-L2 error-tagged corpora in two different languages, one that comes with manually validated UD annotation and one that was only parsed automatically. In both cases, we randomly selected 100 sentences to be used during development and set the rest aside for testing. Carrying out a systematic evaluation was not possible: more often than not, an error tag maps not to a single L1-L2 query, but to a potentially rather large set of queries whose exhaustiveness is hard to verify. As a consequence, we opted for computing the sentence-level precision and recall obtained upon running, for each corpus, a

---

[8]The pattern replacement language is, in many ways, analogous to the pattern replacement language and documented alongside it: github.com/ GrammaticalFramework/gf-ud/blob/master/ doc/patterns.md (last access: 05.04.2023).

single-token, a tree and a sequence example query, all chosen to be descriptive of an error typical of the language at hand. To automate evaluation as much as possible, we also tried to make each query match one of the error labels of the dataset at hand. In this way, performance for a given query can be assessed by simply comparing the sentence IDs returned by the engine with those of the sentences marked with the corresponding error label. Finding exact correspondences was feasible for single-token queries, but challenging for tree and sequence patterns, which tend to be finer-grained. In such cases, we formulated queries describing a subset of the error cases denoted by a certain label and manually inspected sentences marked with it to select the relevant items. By comparing the results obtained on the two corpora, we also aim at getting insights about the ways in which automatic annotation affects the performance of the engine, even though we cannot quantify its impact.

## 4.1 Experiments on manually validated data

### 4.1.1 Data

Our first treebank of choice is the 398-sentence manually validated subset of the VALICO-UD corpus (Di Nuovo et al., 2022), consisting of texts written by Italian L2 learners with various L1 backgrounds. While much smaller than the above mentioned ESL treebank, it was deemed preferable due to its more complete UD annotation.[9]

Error tagging, present as sentence metadata, is XML-like and based on Nicholls (2003), where each label consists of a two-letter code, with the first character representing the general class of error (inflection, omission etc.), and the second generally specifying the POS tag of the word(s) involved. In some cases, VALICO-UD labels also present a third letter, usually denoting an incorrect inflectional feature.[10] In the error tag IDG, for instance, the three letters stand for "Inflection", "Determiner" and "Gender" and are meant to enclose determiners incorrectly inflected for gender.

| | Precision | Recall |
|---|---|---|
| $V_1$ | 43% (40%) | 100% |
| $V_1'$ | 100% (90%) | 100% (64%) |
| $V_2$ | 100% | 40% |
| $V_3$ | - | 0% |
| $V_3'$ | 100% | 100% |

Table 1: Precision and recall of the example queries run on the VALICO-UD corpus. Values in parentheses do not take error annotation issues into account.

### 4.1.2 Queries

Gender being a notorious source of confusion for learners of Italian, we chose the set of two L1-L2 patterns equivalent to IDG as our first test query:

```
V1: AND [
       POS "DET",
       FEATS_ "Gender={$A → $B}"]
```

A second, more complex query is

```
V2: TREE
       (POS "NOUN")
    [ {DEPREL "det", → }
      DEPREL "det:poss" ]
```

which denotes a subclass of the MD (Missing Determiner) VALICO category describing the common error pattern for which the definite article that should precede a possessive modifying a noun is omitted (consider, for instance, the nominal phrase {il → _ *} suo naso - "his nose").

Word order is relatively free in Italian, and finding recurrent patterns in such a small corpus is not easy. Instead, we use a sequence pattern to find particular occurrences of RD (Replacement of Determiner) errors:[11]

```
V3: SEQUENCE [
       LEMMA "non",
       LEMMA "ci",
       LEMMA "essere",
       LEMMA {"nessun*" −> "un*"}]
```

This pattern matches phrases that translate to "there is/are no $x$". In Italian, this is usually expressed with a double negation: not only is there an initial negation, *non*, but the determiner introducing $x$ (*nessuno* or *nessuna*, depending on $x$'s gender) also has negative polarity. However, since this is not the case in most other languages, it is common for L2 speakers to simply use the indefinite article (*un/un'/uno/una*).

### 4.1.3 Results

As displayed in Table 1, recall is perfect for the first query. The low precision should not mis-

---

[9]Due to licensing issues, the UD annotation of the ESL corpus is released separately from the learner essays themselves. Consequently, in order to prevent the text from being reconstructed from the annotation, the LEMMA and FEATS fields are left blank.

[10]An exhaustive description of the error annotation guidelines is given at raw.githubusercontent.com/ElisaDiNuovo/VALICO-UD_guidelines/main/Error_Annotation_Guidelines_v.1.1.pdf (last access: 05.04.2023).

[11]Even though UD patterns do not support general regular expressions, an asterisk at the beginning or end of a string can be used as a wildcard.

lead: one of the false positives is due to an inconsistency in the error annotation and 8 of the remaining 20 false positives are due to cascading errors. It is often the case, in fact, that the gender of the noun the determiner is referred to is wrong, and the incorrect inflection of the determiner introducing it is merely a consequence of that. In this case, the incorrect noun is marked with the RN (Replace Noun) label in case of a lexical error (cf. *la panca* → *il banco\**, "the bench → the desk") or with the ING (Incorrect Noun Gender) label if the noun is incorrectly inflected (cf. *gli uccelli* → *le uccelle\**", "the birds"), while the determiner gets the IDGcascade tag. To avoid matching cascading errors, we can turn $V_1$ into a TREE query and check whether the determiner's gender agrees with the noun's:

```
V'₁: TREE_
     (AND [POS "NOUN",
           FEATS_ "Gender=$A"])
     [AND [POS "DET",
           FEATS_ "Gender={$A → $B}"]]
```

As Table 1 shows, the precision for $V'_1$ is significantly higher. While recall appears to decline, all of the false negatives can be traced back to errors, inconsistencies or incompletenesses in either UD annotation or error tagging (see Appendix B for a complete list of issues found in the VALICO-UD corpus).

When it comes to $V_2$, there are no false positives, while the 3 false negatives are due to alignment errors. In every case, the sentence at hand presents several errors, so that the L1 and L2 trees differ significantly, increasing the difficulty of the alignment task.

$V_3$, on the other hand, only has one expected hit, the sentence -*Non c'è {**nessun** bacio → **una** baciata\*} per me,- ha pensato tristemente.* (-There's no kiss for me,- (s)he thought sadly."), but no matches. Again, the problem seems to be an alignment error, since the expected sentence id is indeed returned if we use a similar L2-only query:

```
V'₃: SEQUENCE [
       LEMMA "non",
       LEMMA "ci",
       LEMMA "essere",
       LEMMA "un*"]
```

## 4.2 Experiments on parsed data

### 4.2.1 Data

To evaluate the tool on automatically annotated data, we used a 2087-sentence subset of the DaLAJ corpus (Volodina et al., 2021).[12] Such corpus is composed of L1-L2 sentence pairs automatically derived from the error-annotated learner corpus of anonymized L2 Swedish essays SweLL (Volodina et al., 2019). More specifically, SweLL essays are processed so that the L2 component of each sentence pair in the DaLAJ corpus contains exactly one morphological or syntactical error. Arguably, this makes automatically parsing the L2 sentences and aligning them to their L1 counterparts significantly easier than it would be if multiple, possibly cascading and/or overlapping errors coexisted. Evaluating the tool on the SweLL corpus itself, however, would have been extremely impractical, as the original versions of the essays are not sentence-aligned.

In terms of error-annotation, since DaLAJ entries only contain one error each, sentence pairs are simply assigned a SweLL error label. SweLL's error taxonomy, thurughly described by Rudebeck and Sundberg (2021), is a two-level classification: error labels are composed of a capital letter, indicating the error's macro-category (Ortography, Lexicon, Morphology, Syntax or Punctuation), followed by a secondary label giving additional information about the type of error and/or the POS involved. The M-Case label, for instance, indicates the presence of a morphological error that has to do with the case inflection of a noun or pronoun.

DaLAJ sentences were parsed with UDPipe 1 (Straka et al., 2016) using the `swedish-talbanken-2.5` model. While not state-of-the-art, UDPipe 1 was preferred over alternatives with higher reported performance due to its speed and ease of use. The resulting dataset is available at `github.com/harisont/L1-L2-DaLAJ`.[13]

### 4.2.2 Queries

We mentioned the M-Case label, used to mark incorrectly inflected nouns and pronouns. Such a label can be mapped to a rather straightforward single-token query:

```
D₁: FEATS_ "Case={$A → $B}"
```

In Swedish, nouns have a definite and an indefinite form. The correct use of these two forms is

---

[12] The preliminary version of the corpus presented in the paper is exclusively composed of sentences presenting lexical errors. In this work, we used a more recent, soon-to-be-released one also covering morphosyntactical errors.

[13] Last access 05.04.2023.

|       | Precision   | Recall |
|-------|-------------|--------|
| $D_1$ | 77% (76%)   | 58%    |
| $D_2$ | 75%         | 90%    |
| $D_3$ | 89%         | 62%    |

Table 2: Precision and recall of the three example queries run on the DaLAJ corpus. Values in parentheses do not take error annotation issues into account.

typically difficult to aquire for L2 learners. As an example of tree query, we therefore use

```
D₂: TREE_
      (FEATS_ "Definite={Def → Ind}")
    [AND [DEPREL_ "det",
          FEATS_ "Definite=Def"]]
```

This denotes sentences where a nominal (typically a noun) is in its indefinite form despite being introduced by a definite determiner (typically an article). In terms of SweLL error labels, these cases are a fraction of those marked as M-Def, which is used to indicate a wide variety of errors concerning definiteness (adjective-noun agreement, missing determiners etc.).

Finally, along the lines of the sequence patterns discussed in Section 3.1.2, we use

```
D₃: SEQUENCE [
      DEPREL_ "advmod",
      {OR [POS "VERB", POS "AUX"],
       DEPREL_ "nsubj" →
       DEPREL_ "nsubj",
       OR [POS "VERB", POS "AUX"]}]
```

to look for sentences where the V2 order is violated following an adverb or an adverbial clause.[14] With this pattern, we cover some of the errors labelled as S-FinV, namely those involving the misplacement of a finite verb.

### 4.2.3 Results

As Table 2 suggests, precision is reasonably good even on our automatically annotated data, while recall fluctuates depending on the query.

When it comes to $D_1$, false negatives are in the almost totality of cases due to the fact that the parser, as it is to be expected, asssigns tokens different dependency labels based on their case (typically, subjects incorrectly inflected in their accusative form are labelled as direct objects and objects in the nominative form become obliques). The vast majority of the false positives is also due not to the query engine itself, but to incorrect alignments deriving from parse errors. In 11 out of 13 such cases, false positives are sentences containing a syntactical error,

---

[14]Note that, for simplicity, we are only looking for sequences where the verb is contiguous to the subject.

which seems to confirm the intuition that nonstandard syntax causes the parser to annotate the sentences incorrectly. In only one case a false positive is due to a wrongly assigned error label. More interesting are the cases in which tokens are correctly aligned, but the correction of a syntax error consists in a rephrasing that happens to also alter the case of one of the words involved, such as in the L2 phrase *Rollerna för barn* (literally "The roles for the children"), corrected as *Barnens roller* ("The children's roles"), transforming the nominative *barn* into a genitive *barnens*.

The tree query, more specific, has only 10 expected matches, allowing for a thorough error analysis. The only false negative seems to be due to an alignment error whose cause is hard to pinpoint. Of three false positives, one derives from incorrect morphological annotation, one from a rephrasing that creates problems at the alignment stage and one from the presence, in the L2 sentence *De ligger på första plats i den ligan!* ("They are in first place in that league!"), of the English word "league", annotated (arguably correctly) as an indefinite but translated, in the correction, to the definite *ligan*.

The relatively low recall for $D_3$ is easily explained by the fact that, as we already discussed, sentences containing syntactical errors are especially challenging for the parser.

## 5   Conclusions and future work

We presented a query engine for L1-L2 parallel UD treebanks, the first in a larger collection of tools for L2 UD treebanks. The tool, which does not rely on an underlying error taxonomy, allows to search for error-correction pairs via L1-L2 patterns, i.e. pairs of morphosyntactical structures expressed in a pattern matching language for dependency trees, which we extended in order to simplify its use on parallel treebanks. Our novel retrieval algorithm allows searching for full sentences as well as extracting their specific query-matching portions.

Our first, small-scale evaluation of the tool gives promising results, but also shows that the alignment component is often the bottleneck. This propmts us to investigate alignment techniques specifically meant for parallel learner corpora, such as Felice et al. (2016)'s for L2 English. The fact that, for automatically annotated data, many alignment issues derive from parse errors also

seems to confirm the scarce robustness of standard tools to learner errors, pointing to a need to train *ad hoc* models or explore new, more specific approaches.

In future versions of the tool, we plan to optimize variables and generalize them to all UD fields, thus increasing expressive power of the query language. Furthermore, while the tool is designed with L1-L2 treebanks in mind, nothing prevents us from testing it on multilingual parallel UD treebanks, for example to find instances of known translation divergences.

As for the future of L2-UD, our efforts in the near future will be focused on extracting error patterns from L1-L2 treebanks. Eventually, we hope it will also be possible to integrate the two and enable using error-correction pairs to retrieve similar examples. This would help making the engine more user-friendly, replacing explicit queries, but could also be a strategy to provide L2 learners with feedback, along the lines of Arai et al. (2019).

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

## Appendix A  Abbreviations

### A.1  UD standard

### A.2  CoNNL-U fields

- DEPREL: dependency label;
- FEATS: list of morphological features;
- FORM: word form or punctuation symbol;
- LEMMA: lemma/stem of the word form;
- POS: Universal POS tag;
- XPOS: language-specific POS tag.

For the full specification of the CoNNL-U format, see `universaldependencies.org/format.html` (last access 05.04.2023).

### A.2.1  Universal POS tags

- ADP: adposition (pre- or postposition);
- ADV: adverb;
- AUX: auxiliary;
- DET: determiner;
- PRON: pronoun;
- VERB: non-auxiliary verb.

For a comprehensive list of UD POS tags, see `universaldependencies.org/u/pos` (last access 05.04.2023).

### A.2.2  Universal dependency relations

- advmod: adverbial modifier of a predicate or modifier word;
- amod: adjectival modifier of a nominal;
- aux: auxiliary;
- case: case-marking element treated as a separate word;
- det: determiner. The subtype poss indicates a possessive;
- nsubj: nominal subject;
- obj: direct object;
- obl: oblique nominal, i.e. non-core verb argument or adjunct;
- root: root of the sentence, usually its main (non-auxiliary) verb and, in general, a content word.

For a comprehensive list of UD relations, see `universaldependencies.org/u/dep` (last access 05.04.2023).

### A.3  VALICO-UD error labels

- IDG: Determiner incorrectly Inflected for Gender. The IDGcascade label is used in cases where the incorrect inflection depends on another error, typically ING or RN;
- ING: Noun incorrectly Inflected for Gender;
- RD: Determiner Replacement (wrong choice of determiner);
- RN: Noun Replacement (lexical error involving a noun);
- SEU: Spelling error - Unnecessary apostrophE.

The complete error annotation guidelines for the VALICO-UD treebank are available at `raw.githubusercontent.`

### A.4 SweLL error labels

- M-Case: noun or pronoun incorrectly inflected for case;
- M-Def: definiteness-related error, namely either:
    - noun, pronoun, adjective or participle incorrectly inflected for definiteness;
    - incorrect, missing or redundant article;
- S-FinV: incorrect placement of a finite verb.

See Rudebeck and Sundberg (2021) for the full SweLL correction annotation guidelines.

## Appendix B    Annotation inconsistencies

### B.1    VALICO-UD corpus

- Sentence 34-12_en-3: *{un → un'*}* correctly labelled as SEU, but not as IDG[15]
- sentence 19-06_en-3 *un* in *{un uomo → un'uoma*}* labelled as IDG rather than IDG-cascade
- sentence 18-10_en-1 *la* in *{sul braccio → sul+la braccia*}* labelled as IDG rather than IDGcascade
- sentence 18-05_en-1 token *sedile* lacking gender annotation in both the L1 and the L2 files
- sentece 17-07_en-2 token *amante* lacking gender annotation in both the L1 and the L2 files
- sentece 3-13_fr-3 *le* in *{gli uccelli → le uccele*}* labelled as IDG rather than IDGcascade.

This list of error annotation issues refers to the 14.05.2022 version of the treebank.[16] At the time of writing, these observations have been discussed

with the authors of the treebank and the annotations in question, excepts for two cases in which they were the result of a deliberate choice, are in the process of being fixed.

### B.2    DaLAJ corpus

- Sentence 1811 labelled as S-Clause rather than M-Case.

This annotation error refers to the 04.02.2023 version of the treebank and was corrected in a subsequent update.[17]

---

[15] The sentence in question is *Ho voltato la pagina e ho iniziato a leggere {un → un'*} altro titolo.* ("I turned the page and started reading another title."). This is an interesting case: the UD annotation correctly states that the indefinite article *un* is masculine while its L2 counterpart *un'* (mind the apostrophe) is feminine, but the manually assigned error tag, rather than the expected IDG, is SEU, which indicates, also correctly, a spelling error (unnecessary apostrophe). All the more reasons not to rely on explicit error labelling!

[16] Available for download at github.com/UniversalDependencies/UD_Italian-Valico (last access 05.04.2023), with commit SHA 7c4fae4f1e6491ca9e648cfb902e1c675c179a42.

[17] Available for download at github.com/harisont/L1-L2-DaLAJ (last access 05.04.2023), with commit SHA 94e133aa083e487cfb28a7c22dda4e1c240bcaf5.

# Appendix C   Example program output: Markdown reports

The following reports, as well as all results presented in this paper, were obtained with L2-UD v0.[18]

## C.1  `TREE_ (AND [POS "NOUN", FEATS_ "Gender=$A"]) [AND [POS "DET", FEATS_ "Gender=$A → $B"]]` $(V_1')$

**Sentence 30-09_de-2:**

| L1 sentece | L2 sentece |
| --- | --- |
| Poi , lei si è arrabbiata e mi ha detto che questo uomo con i grandi muscoli che si è sdraiato a terra era il suo fidanzato e **il suo** grande **amore** . | Poi , lei si è arrabbiata e mi ha detto che questo uomo con i grandi muscoli che si è sdraiato sulla terra era il suo fidanzato e **la sua** grande **amore** . |

**Sentence 3-11_fr-3:**

| L1 sentece | L2 sentece |
| --- | --- |
| La donna ringraziava il suo salvatore con un abbraccio e chiudeva **gli occhi** . | La dona ringraziava suo salvatore con un braccio e chiusa **le occhi** . |

**Sentence 10-07_es-1:**

| L1 sentece | L2 sentece |
| --- | --- |
| Gli ho gridato **alcune parolacce** . | L' ho gridato **alquini parolace** . |

**Sentence 4-04_fr-2:**

| L1 sentece | L2 sentece |
| --- | --- |
| Un **altro uomo** si trovava lì , seduto su una panchina del di il parco , leggendo un giornale con i suoi occhiali . | Un **altra uomo** , si trova li , seduto sul su il un panchino del di il parco , leggendo un giornale con i suoi occhiali . |

**Sentence 34-12_en-3:**

| L1 sentece | L2 sentece |
| --- | --- |
| Ho voltato la pagina e ho iniziato a leggere **un** altro **titolo** . | Ho voltato la pagina e ho iniziato a leggere **un'** altro **titolo** . |

**Sentence 19-01_en-3:**

| L1 sentece | L2 sentece |
| --- | --- |
| Ieri al a il parco , **un uomo** brutto è arrivato e ha detto delle parole cattive a una donna . | Ieri al a il parco , **un' uomo** brutto è arrivata e ha detto le parole cattive a una donna . |

**Sentence 27-02_de-3:**

---

[18]Download link: `github.com/harisont/L2-UD/releases/tag/v0` (last access 05.04.2023).

| L1 sentece | L2 sentece |
| --- | --- |
| Subito guarda come un altro uomo con grande forza fisica e con malumore porta sulle **sue spalle** una ragazza che grida . | Subito guarda come un altro uomo con grande forza fisica e con malumore porta una ragazza sulle **sui spalle** che grida . |

**Sentence 3-12_fr-3:**

| L1 sentece | L2 sentece |
| --- | --- |
| Era **un** vero **momento** di benessere . | Era **una** vero **momento** di benessere . |

**Sentence 27-08_de-3:**

| L1 sentece | L2 sentece |
| --- | --- |
| « Il **mio amore** non dipende dal suo comportamento . » | « Il **mia amore** non dipende dal suo comportamento . » |

**Sentence 32-06_de-2:**

| L1 sentece | L2 sentece |
| --- | --- |
| Un uomo molto forte , intelligente , sportivo e carino mi ha salvato da **questa** ignorante **persona** , Marco . | Un' uomo molto forte , intelligente , sportivo e carino mi ha salvato di **questo** ignorante **persona** Marco . |
| **Un uomo** molto forte , intelligente , sportivo e carino mi ha salvato da questa ignorante persona , Marco . | **Un'** **uomo** molto forte , intelligente , sportivo e carino mi ha salvato di questo ignorante persona Marco . |

**C.2  SEQUENCE [DEPREL_ "advmod", OR [POS "VERB", POS "AUX"], DEPREL_ "nsubj" -> DEPREL_ "nsubj", OR [POS "VERB", POS "AUX"]]** ($D_3$)

**Sentence 1958:**

| L1 sentece | L2 sentece |
| --- | --- |
| **Därför tycker jag** om havet . | **Därför jag tycker** om havet . |

**Sentence 1943:**

| L1 sentece | L2 sentece |
| --- | --- |
| **Tyvärr har någonting hänt** som gör att jag inte kan gå på kursen och jag önskar att få pengarna tillbaka . | **Tyvärr någonting har hänt** som gör att jag inte kan gå på kursen och jag önskar att få pengarna tillbaka . |

**Sentence 1950:**

| L1 sentece | L2 sentece |
| --- | --- |
| **Därför vill** jag inte **flytta** . | **Därför jag vill** inte **flytta** . |

**Sentence 1965:**

| L1 sentece | L2 sentece |
| --- | --- |
| **Där bodde jag** i Göteborg med min mamma och hennes hundar i hennes hus . | **Där jag bodde** i Göteborg med min mamma och hennes hundar i hennes hus . |

**Sentence 1936:**

| L1 sentece | L2 sentece |
| --- | --- |
| Ibland **kan vi titta** och lyssna på hur det funkar . | Ibland **vi kan titta** och lyssna på hur det funkar . |

**Sentence 1139:**

| L1 sentece | L2 sentece |
| --- | --- |
| **Därför** har **man kommit** med förslaget att ha en kurs angående arbetslivet i gymnasiet . | **Därför man kommit** med förslaget att ha en kurs angående arbetslivet i gymnasiet . |

**Sentence 1942:**

| L1 sentece | L2 sentece |
| --- | --- |
| Lyckligtvis kan du spela piano , och **där** kan **du lära** känna nya vänner . | Lyckligtvis kan du spela piano , och **där du** kan **lära** känna nya vänner . |

**Sentence 1976:**

| L1 sentece | L2 sentece |
| --- | --- |
| **Ibland brukar man** säga att kärleken har ingen gräns och det är sant . | **Ibland man brukar** säga att kärleken har ingen gräns och det är sant . |

**Sentence 1939:**

| L1 sentece | L2 sentece |
| --- | --- |
| Men nu **är jag** inte **intresserad** av den längre . | Men nu **jag är** inte **intresserad** av den längre . |