# OpenReview forum: "A query engine for L1-L2 parallel dependency treebanks"
_NoDaLiDa/2023/Conference — NoDaLiDa 2023_

### Official Review · Reviewer_1YkD · 2023-03-01
**the paper presents an application and extension of the GF-UD query language for studying L2 learner corpora**

**Rating:** 6
**Confidence:** 4

**Review:**

The paper presents a search application for L2 learner corpora, where utterances of the learner (containing errors) are paired with correct sentences. The application domain appears to be very specialized (ie not just L2 data, but parallel L2 data), which somewhat limits the relevance of the study.

Positive points of the paper are the fact that it contains some discussion of this type of L2 corpora in general, and argues for UD as uniform annotation layer, without the need for application specific additional mark-up (of errors).

The GF-UD query language seems user-friendly and expressive enough for a range of queries over UD treebanks.

The alignment mechanism outlined in section 3.2 (sentence retrieval) seems similar to what is being used for parallel corpora in MT, where there are a number of tools an algorithms for doing cross-lingual word alignment.

The GF-UD query language appears to be a query language of trees. Grew-match is currently perhaps the most popular query language for UD treebanks (http://universal.grew.fr) This query language is based on a generic query language for graphs. It would be interesting to explore to what extent the current application can also be modeled as a graph search problem (with the two sentences and their word-level alignment forming a graph).

The experiments use a small corpus and an as-yet unreleased corpus. This underlines the fact that the relevance of the application might be limited to a small user-base and dataset.

**Paper Type:**

Short paper

---

### Official Review · Reviewer_vYPs · 2023-03-09
**Clear and straightforward description of novel tool**

**Rating:** 7
**Confidence:** 3

**Review:**

The authors introduce a querying tool for L1-L2 treebanks and demonstrate its use and capabilities on a number of practical examples.

The paper is well-written, with a clearly stated goal, straightforward execution, and thorough description of use-cases. Particularly commendable is the thorough error analysis for example queries. Even though the use-case is very highly specific, I consider this paper a strong contribution and see no technical drawbacks or issues with the methodology.


Typos and minor errors:

657: in every case[s]

725: such [a] label

**Paper Type:**

Long paper

---

### Decision · Program_Chairs · 2023-03-17

Accept